# Reverse Engineering of the Pediatric Sepsis Regulatory Network and Identification of Master Regulators

**DOI:** 10.3390/biomedicines9101297

**Published:** 2021-09-23

**Authors:** Raffael Azevedo de Carvalho Oliveira, Danilo Oliveira Imparato, Vítor Gabriel Saldanha Fernandes, João Vitor Ferreira Cavalcante, Ricardo D’Oliveira Albanus, Rodrigo Juliani Siqueira Dalmolin

**Affiliations:** 1Bioinformatics Multidisciplinary Environment–BioME, Instituto Metrópole Digital, Universidade Federal do Rio Grande do Norte, Natal 59078-400, Brazil; raffael.azevedo@gmail.com (R.A.d.C.O.); xdanilo@ufrn.edu.br (D.O.I.); vitor.saldanha.095@ufrn.edu.br (V.G.S.F.); jvfecav@gmail.com (J.V.F.C.); 2Department of Computational Medicine & Bioinformatics, University of Michigan, Ann Arbor, MI 48109, USA; albanus@umich.edu; 3Department of Biochemistry–DBQ–CB, Federal University of Rio Grande do Norte, Natal 59064-741, Brazil

**Keywords:** master regulators, pediatric sepsis, regulatory networks, systems biology, septic shock

## Abstract

Sepsis remains a leading cause of death in ICUs all over the world, with pediatric sepsis accounting for a high percentage of mortality in pediatric ICUs. Its complexity makes it difficult to establish a consensus on genetic biomarkers and therapeutic targets. A promising strategy is to investigate the regulatory mechanisms involved in sepsis progression, but there are few studies regarding gene regulation in sepsis. This work aimed to reconstruct the sepsis regulatory network and identify transcription factors (TFs) driving transcriptional states, which we refer to here as master regulators. We used public gene expression datasets to infer the co-expression network associated with sepsis in a retrospective study. We identified a set of 15 TFs as potential master regulators of pediatric sepsis, which were divided into two main clusters. The first cluster corresponded to TFs with decreased activity in pediatric sepsis, and *GATA3* and *RORA*, as well as other TFs previously implicated in the context of inflammatory response. The second cluster corresponded to TFs with increased activity in pediatric sepsis and was composed of *TRIM25*, *RFX2*, and *MEF2A*, genes not previously described as acting in a coordinated way in pediatric sepsis. Altogether, these results show how a subset of master regulators TF can drive pathological transcriptional states, with implications for sepsis biology and treatment.

## 1. Introduction

Sepsis is a syndrome of physiologic and biochemical abnormalities in response to an infection, which leads to systemic damage across organs and tissues [1]. Pediatric sepsis has distinct clinical presentations compared to adult sepsis regarding the prevalence of cardiovascular dysfunction, acute respiratory distress syndrome, and organ failure which are more likely to happen in children compared to adults [2]. In children, infections represent about 40% of hospitalizations, and almost five million children progress to severe organ dysfunction worldwide related to sepsis [3]. Sepsis is the leading cause of death in intensive care units (ICUs), affecting approximately 30% of all ICU patients worldwide [4]. There is broad variability in sepsis prevalence and outcomes in ICU patients [4,5], as sepsis can manifest different clinical presentations, ranging from systemic inflammatory response syndrome (SIRS), sepsis, septic shock, and severe sepsis [6,7]. Each of these clinical presentations requires a different set of medical interventions. Therefore, understanding the differences across sepsis subtypes is critical to patient outcomes.

Previous studies aimed to stratify septic patients based on clinical and molecular aspects with mixed success [8,9,10], indicating that sepsis stratification could benefit from additional molecular biomarkers. However, there are only a few described molecular approaches to discriminate across sepsis prognostics, and individual-level variations in the immune system can act as confounders [11]. Several gene expression panels aim to identify pediatric sepsis biomarkers, but there is still a lack of consensus about what constitutes the pediatric sepsis gene signature [12]. Overcoming these challenges will lead to higher success in patient stratification and improve treatment outcomes [13].

A promising approach to identify sepsis molecular biomarkers is to investigate transcriptional regulatory mechanisms of sepsis. Currently, there are computational frameworks designed to identify gene regulatory networks. Gene regulatory networks are composed of transcription factors (TFs) and their target genes. Each TF and its target genes are collectively called a regulon, and the subset of TFs statistically associated with the genes perturbed in a given condition (i.e., its transcriptional signature) are called master regulators (MRs). The MR framework is a powerful approach and has been previously applied to cancer and other diseases [14,15,16,17,18,19,20,21]. Previous studies have analyzed gene expression profiles of sepsis and pediatric sepsis [22,23,24,25,26], but none of these have attempted to directly identify the transcriptional MRs of sepsis and pediatric sepsis. Therefore, the MR framework represents a strategy ripe with the potential to uncover novel biological and clinical insights into sepsis.

Here, we reconstructed the pediatric sepsis regulatory network aiming to identify sepsis-associated transcriptional regulators (sepsis MRs). To this end, we applied a previously described robust mutual information framework [27] to publicly available whole-blood high-throughput gene expression molecular profiles from pediatric sepsis patients. One novelty of our approach was to control for transcriptional signatures that are shared across other unrelated inflammatory conditions. Therefore, the MRs identified by our analyses are highly enriched to be specific to sepsis. We identified 15 sepsis transcriptional MRs broadly divided into two non-overlapping regulatory networks that include both known and novel TFs associations with sepsis. Altogether, these results will likely have implications in understanding sepsis biology and treatment.

## 2. Materials and Methods

### 2.1. Pediatric Sepsis Regulatory Network Reconstruction

The complete pipeline for regulatory network reconstruction is depicted in Figure 1 and Appendix A. Data used to reconstruct the pediatric sepsis regulatory networks were retrieved from Gene Expression Omnibus (GEO) [28] (accession numbers GSE13904 and GSE4607). The dataset GSE13904 includes expression data from whole blood samples of children diagnosed with SIRS, sepsis, and septic shock [6]. The dataset GSE4607 includes expression data from whole blood samples of children diagnosed with SIRS, and septic shock [23,24,25,29]. Considering the constant evolution of sepsis definitions, samples collected in the 2000s may not necessarily be described in the same way in 2021. Therefore, we collectively name the case samples as sepsis from now on. The list of 1388 human TFs was obtained from Fletcher2013b R/Bioconductor package (R version 4.1.0) [15].

The sepsis regulatory network reconstruction was performed using RTN (Reconstruction of Transcriptional Networks) [15], an R/Bioconductor package (R version 4.1.0). This tool was designed to assess gene regulatory networks by using mutual information of gene expression data and a predefined set of transcription factors (TFs) to create a TF-centered regulatory map. RTN uses the Algorithm for the Reconstruction of Accurate Cellular Networks (ARACNe) [30] to infer the regulatory networks [15]. The methodology aims to allow reliable reconstruction of genome-wide mammalian networks, overcoming limitations such as temporal gene expression alterations between samples, cellular populations characteristics, and random alterations in expression states of different samples [30]. Briefly, the RTN algorithm identifies statistical dependence of expression data using mutual information along with a set of transcription factors (TFs) to build a TF-centric regulatory network. Each TF and its target genes are collectively called a regulon. The Transcriptional Network Inference (TNI) algorithm calculates the association between a TF and the genes in its regulon (i.e., a potentially regulated gene), and removes spurious links by performing random permutations (10,000 permutations, BH adjusted *p*-value < 0.05). We reconstructed two independent regulatory networks, from now on named Pediatric Sepsis 1 (using GSE13904, number of samples = 99) and Pediatric Sepsis 2 (using GSE4607, number of samples = 108). Each network is depicted based on regulon overlap, and the nodes represent the inferred regulons with at least 15 genes.

Both microarray datasets used for network inference were generated by the same platform (Affymetrix Human Genome U133 Plus 2.0), preprocessed, and normalized using the affy R package (R version 4.1.0) [31]. Annotation for this platform was obtained from the platform annotation file provided in the GEO database.

### 2.2. Gene Signatures from Sepsis, Rheumatoid Arthritis, and Multiple Sclerosis

The goal of a gene signature is to identify genes that are altered in a specific biological phenomenon. Here, the gene signatures consist of a list of differentially expressed genes in a case-control expression experiment. The signature is employed to identify the putative master regulators among the network regulons in the Master Regulator Analysis (MRA), as described below (Section 2.3). The datasets to infer the gene signatures were obtained from GEO. The dataset GSE26378, used to infer sepsis gene signature, contains expression data from children diagnosed with septic shock (sepsis group, number of samples = 82), and healthy children (control group, number of samples = 21) [32]. The dataset GSE56649, used to infer rheumatoid arthritis (RA) gene signature, contains expression data from 22 CD4+ T-cell samples of RA patients (RA group, number of samples = 13), and healthy donors (control group, number of samples = 9) [33]. The dataset GSE21942, used to infer rheumatoid multiple sclerosis (MS) gene signature, contains expression data from 29 peripheral blood mononuclear cells (PBMC) samples of MS patients (MS group, number of samples = 12), and healthy donors (control group, number of samples = 15) [34]. All microarray datasets used to infer the gene signatures were generated by the same platform (Affymetrix Human Genome U133 Plus 2.0), preprocessed, and normalized using the affy R package [31]. Annotation for this platform was obtained from the platform annotation file provided in the GEO database. Differentially expressed genes for each signature were obtained using the limma R/Bioconductor package (R version 4.1.0) [35] in a case-control design (BH adjusted *p*-value < 0.05).

### 2.3. Master Regulator Analysis

Master Regulator Analysis (MRA) uses hypergeometric testing to check whether a list of genes corresponding to a transcriptional signature (e.g., differentially expressed genes in disease vs. control) are significantly enriched in a given regulon. If this test has statistical significance (BH adjusted *p*-value < 0.05), the TF controlling a given regulon is likely to be considered a candidate MR. The MRA pipeline from the RTN R/Bioconductor package was applied in Pediatric Sepsis 1 and 2 datasets. We also interrogated both Networks (Pediatric Sepsis 1 and Pediatric Sepsis 2) with two different inflammatory disease gene signatures for MS (GSE21942) and RA (GSE56649). The objective of this step was to identify among those regulons assigned as MRs using sepsis gene signature, which ones can also be identified as MRs using other inflammatory disease gene signatures. The GSE56649 (used for RA gene signature) and GSE21942 (used for MS gene signature) are not age-matched with both sepsis datasets. Additionally, the sepsis datasets were obtained from whole blood while RA and MS datasets were obtained from PBMC. Therefore, we can use this information to determine which regulons are regulated by sepsis-specific MRs, as TFs associated with regulons detected in datasets from different inflammatory syndromes, patient ages, and blood cell populations, likely represent non-specific associations. Appendix A details the filtering steps to narrow down the number of MR candidates.

### 2.4. Network Visualizations

The RedeR R/Bioconductor package (R version 4.1.0) was used to visualize the graphs generated by the RTN R/Bioconductor package [36]. RedeR allows the visualization of networks in several forms, but here we used association maps (amap) and tree-and-leaf representations. In association maps, nodes represent a regulon, while edges depict mutual regulation of genes by two TFs (i.e., regulatory overlap between regulons). Edge width corresponds to the number of genes mutually regulated. The tree-and-leaf visualization lays the network as a dendrogram in which edges are distributed according to the clustering of regulatory overlaps. By default, regulons with less than 15 genes are not represented.

### 2.5. Regulon Activity Analysis

Regulon activity was assessed using the RTN Bioconductor/R package. The RTN package measures for every patient how the expression of each gene deviates towards the average expression of that gene for all patients in the cohort and then applies the Two-Tailed Gene Set Enrichment Score analysis (GSEA2). The TFs activity level was approximated by the Normalized Enrichment Score (NES) computed for its regulon. This analysis measures the enrichment score of the regulatory activity for each regulon, compared with each other, for a set of patients in a cohort. This measure is basically the deviation of the average expression of a given gene for all patients of the cohort. The results are depicted in a heatmap format, along with dendrograms.

### 2.6. Gene Ontology Functional Enrichment

To investigate the function of regulons, we functionally enriched the regulated gene sets of each MR. This analysis was carried out through the clusterProfiler R/Bioconductor package (R version 4.1.0) [37]. Genes were enriched under the Biological Process category from Gene Ontology (GO). Functional associations with adjusted *p*-values under 0.05 (Benjamini-Hochberg correction) were considered to be significant.

### 2.7. Master Regulator Expression

For MR expression, we used normalized expression of each one of the 15 MR-coding genes. Since more than one probe represents the same gene, the first step was to build a score to select the best probe. This score consists of:*geomSD(P)^1/geoMean(P)^*(1)
where *geomSD* is the geometric standard deviation, *geoMean* is the geometric mean, and *p* is the value for a given probe. For each gene, the function seeks all probes and gets the one with the highest score, representing then the MR-coding gene for a given MR, ranked by adjusted *p*-value of a pairwise Wilcoxon’s test (adjustment by Bonferroni *p* < 0.01). Afterwards, gene expression on case (septic) and control samples was measured for both Pediatric Sepsis 1 (GSE13904, case = 99; control = 18) and Pediatric Sepsis 2 (GSE4607, case = 108; control = 15). Expression comparison between case and control of each network was measured by adjusted *p*-value of pairwise Wilcoxon’s test, by Bonferroni.

## 3. Results

### 3.1. Sepsis Regulatory Networks and Master Regulator Analysis

The main objective of this study was to reconstruct the gene regulatory network of sepsis. Regulatory networks are specific for a given biological condition and show the interactions among regulatory units, also called regulons. A given regulon includes a transcription factor (TF) and its regulated genes. Once the regulatory network is constructed, MRs can be defined by identifying which regulons are significantly enriched with a given gene signature [14]. The networks from Pediatric Sepsis 1 and 2 datasets are depicted in Figure 2 and Figure 3, respectively. The networks are represented by a tree-and-leaf graph in which nodes depict regulons, and edges depict the overlap between regulons in a hierarchical fashion. In other words, the closer the regulons are in the network, the greater their overlap regarding regulated genes. Node size corresponds to the number of genes in the regulon while color indicates adjusted *p*-values enrichment for sepsis gene signature, highlighting the MRs. MRA identified 233 MRs in the Pediatric Sepsis 1 network (Appendix A), and 215 in the Pediatric Sepsis 2 network (Appendix A), of which 179 MRs were present in both networks (Appendix A, Appendix A). Both networks showed two MR clusters, which we named clusters A and B. In the Pediatric Sepsis 1 network, cluster A comprised 39 MRs, and cluster B comprised 34 MRs. In the Pediatric Sepsis 2 network, cluster A comprised 68 MRs, and cluster B comprised 50 MRs. From all the 179 MRs present in both networks, 50 were contained in these clusters (Appendix A, Appendix A). As expected from these early steps of the analyses, the putative MRs included not only TFs known to be associated with the immune response (e.g., *IRF4* and *GATA3*), but also others that may represent lineage-specific regulons (e.g., *KLF* families of TF) or gene activators, such as TFs forming the AP-1 complex (*FOS* and *JUNB*). This indicated that our MR analysis was also prioritizing TFs that were not specific to sepsis. We reasoned that this lack of specificity was due to the septic shock transcriptional signature including genes that were not directly linked to sepsis, but rather resulting from downstream perturbations.

As our previous results suggested that the candidate MR list included TFs not associated with sepsis, we next sought to identify and remove non-specific TFs from the putative MR list. To this end, we then performed MRA in both networks (Pediatric Sepsis 1 and 2) using two unrelated inflammatory disease gene signatures (multiple sclerosis and rheumatoid arthritis). Appendix A shows the intersection between all gene signatures (sepsis, RA, and MS). Among the 50 MRs inside the clusters of both Pediatric Sepsis Network 1 and 2, we observed a subset of 15 MRs that were specifically enriched only in the sepsis signature, and therefore likely represent true MRs of the transcriptional states associated with sepsis. The prioritized MRs were *GATA3*, *HOXB2*, *KLF12*, *MEF2A*, *NR3C2*, *RFX2*, *RORA*, *TRIM25*, *ZKSCAN8*, *ZNF134*, *ZNF234*, *ZNF235*, *ZNF329*, *ZNF331*, and *ZNF529*. Appendix A show the MRA of the 15 MRs in both Pediatric Sepsis 1 and Sepsis 2 networks, respectively. From this point on, the analysis focused on these 15 MRs (see Appendix A to gene aliases and protein name of the 15 MRs). In the Pediatric Sepsis 1 network, 11 out of 15 MRs (*GATA3*, *HOXB3*, *KLF12*, *NR3C2*, *RORA*, *ZKSCAN8*, *ZNF234*, *ZNF235*, *ZNF329*, *ZNF331*, and *ZNF529*) are located in cluster A (Figure 2). In the Sepsis 2 network, 12 out of 15 MRs (*GATA3*, *HOXB3*, *KLF12*, *NR3C2*, *RORA*, *ZKSCAN8*, *ZNF134*, *ZNF234*, *ZNF235*, *ZNF329*, *ZNF331*, and *ZNF529*) are also located in cluster A (Figure 3). Notably, these MRs formed similar connectivity patterns across these two independently generated networks, which highlights the robustness of our computational approach and is consistent with these MRs performing similar and complementary biological functions [14].

### 3.2. Master Regulator Activity

Regulon activity was assessed based on the expression of genes in the regulon. Briefly, gene expression was first converted into a z-score and, in each sample, all genes were sorted by the z-score and then used as the reference list in GSEA2. A given regulon is considered activated when its positively regulated genes were up-regulated in the GSEA, and its negatively regulated genes were down-regulated in the GSEA. Conversely, the regulon was considered inhibited when its negatively regulated genes were up-regulated in the GSEA, and positively regulated genes were down-regulated in the GSEA [17]. Figure 4A shows the heatmaps of the final set of 15 MRs, showing a cluster formed by *RFX2*, *MEF2A*, and *TRIM25* in both heatmaps. These three MRs show an inverse pattern of activity when compared to the other 12 MRs. Figure 4B shows the functional overlap among the regulons, based on the proportion of genes mutually regulated by each regulon pair. Similar to what was observed in the heatmaps, *RFX2*, *MEF2A*, and *TRIM25* regulons cluster together (depicted as green nodes in the network).

### 3.3. Regulon Similarity

We assessed the similarity between regulons. Briefly, we calculated how many genes are regulated in common between the two networks. Figure 4C shows the number of shared genes among the regulons for both Pediatric Sepsis 1 network and Pediatric Sepsis 2. The most similar regulon comparing both networks is *GATA3*, and the least similar is *ZNF134*. It is noteworthy that the top four MRs ranked the same way in both datasets (Appendix A), but only *GATA3*, *KLF12*, and *ZNF529* also were the top MRs in shared regulon size. Again, this highlights the robustness and reproducibility of our approach and indicates that these top-scoring TFs are likely the true MRs driving the pediatric sepsis transcriptional signature.

### 3.4. Regulon Functional Enrichment

We next sought to determine what processes were associated with the prioritized regulons. To this end, we performed the functional enrichment of regulons. As the regulons were highly concordant across both networks, we merged regulons from both networks to increase statistical power. Figure 5 shows enrichment results according to GO Biological Process (BP) ontology. Only four regulons were not enriched due to the statistical cut-off. *GATA3* regulated genes are notably related to non-coding RNAs (such as tRNAs and other non-coding RNAs, such as miRNAs and lcRNAs) and lymphocyte activation. The latter result is consistent with GATA3, specifically driving the sepsis signature in T cells, consistent with previous literature [38]. Some regulons also had a similar enrichment pattern, such as *KLF12*, *ZNF234*, *ZNF23*, and *ZNF529* (related to ncRNAs); *RFX2* and *RORA* showed overlap regarding lymphocyte-related activities. *MEF2A* was the only regulon enriched for apoptosis-related BP terms. The other regulons did not notably share biological processes. *NR3C2* was enriched in cell cycle arrest, while *TRIM25* was enriched for neutrophil activation and erythrocyte maturation, and *ZNF331* and *ZNF529* was enriched for DNA replication processes. This suggests that these TFs regulate the pediatric sepsis transcriptional signature acting through distinct pathways. When taken together, these results highlight the complexity of the pediatric sepsis transcriptional signature, which depends on the activation of TFs with distinct regulatory profiles and propagates itself across multiple biological pathways and cell types.

### 3.5. Master Regulators Expression in Sepsis Datasets

The expression of the 15 MRs was accessed in both Pediatric Sepsis 1 and 2 datasets by comparing the septic samples and healthy controls. Figure 6 shows the gene expression of each MR candidate in both datasets. The first row of boxplots shows that *MEF2A*, *RFX2*, and *TRIM25* were more expressed in sepsis samples in contrast to control ones. Two of the remaining 12 MRs showed no significant difference in expression, *ZNF235* and *ZKSCAN8*, with a false discovery rate (FDR) < 0.01, using a pairwise Wilcoxon’s test. The remaining 10 MRs showed an inverse expression pattern, being more expressed in control samples when compared to septic ones. This result independently supports our previous MR activity clustering. All downregulated MRs in pediatric sepsis belonged to Cluster A, indicating that sepsis progression involves decreased activation of this subset of TFs. On the other hand, all MRs belonging to Cluster B (*MEF2A*, *RFX2*, and *TRIM25*) had increased activity. To our knowledge, this is the first time these TFs are directly implicated in pediatric sepsis progression. Therefore, these candidate MRs may represent novel insights in pediatric sepsis biology and are ripe targets for exploration in future clinical studies. An important additional interpretation of these results is that the changes in gene expression associated with the candidate MRs had small effect sizes, which would likely be missed in differential gene expression analyses. This highlights the power of the MR approach to detect key drivers of disease states.

## 4. Discussion

Pediatric sepsis is a genuine Pandora’s box regarding its complexity. It is a challenging acute syndrome that quickly evolves into life-threatening scenarios and the establishment of new reliable prognostic and predictive biomarkers remains necessary [12]. Identifying regulatory mechanisms of this clinical condition could help the search for specific traits indicating disease status [39]. The natural variation within individuals may lead to different outcomes despite identical protocols and treatments [40]. Symptoms of pediatric sepsis are often dependent on subject age, sepsis type, sepsis duration, and etiological agent. Here, we used expression data from two independent cohorts to reconstruct the pediatric sepsis regulatory networks. We identified the same 15 MRs in both cohorts, the composition of those MRs were similar in both cohorts, as well as the MRs activity, and the MRs expression in pediatric septic patients. There are hundreds of biomarkers associated with sepsis, but very few (e.g., C reactive protein and procalcitonin) are currently used as a clinical routine, indicating that we might lack knowledge about how sepsis regulates gene expression [41]. Previous studies using the same datasets used here identified alterations in zinc ion homeostasis and enhanced interleukin-8 and metallothioneins (MT) expression [6]. Authors also identified *IL-27* as a diagnostic biomarker for bacterial infection in critically ill children, which was further validated in independent datasets [23,24,25,29].

From the 15 pediatric sepsis MRs, 12 cluster together in the same regulatory network branch. This proximity indicates that these 12 MRs acts coordinately and share the regulations of target genes. Downregulation of *GATA3* in septic patients is related to Th1 cell differentiation, associated with immunosuppressed states [38]. The Th2/Th1 immune response imbalance with Th2 dominance was associated with a bad prognosis [42]. *RORA* regulates genes in the circadian cycle [43], and a healthy circadian cycle positively impacts sepsis survival rate [44]. *NR3C2* is down-regulated in septic patients compared to healthy subjects. This gene encodes a receptor for mineralocorticoids and glucocorticoids, such as aldosterone and cortisol, respectively [45,46]. Cortisol is known to be a potent anti-inflammatory, and the *NR3C2* downregulation might be a crucial step in disease progression [47]. We identify steroid administration as a source of expression variability in the dataset GSE4607, used to construct Sepsis network 2 (see Covariate selection in Appendix A). It would be naive to assume that these 15 MRs are uniquely responsible for post-inflammatory events in sepsis. It is clear that a plethora of other TFs, identified as MRs, are also responsible for the different septic outcomes; however, the stringency of the analysis assures that these 15 MRs are likely to be directly involved in sepsis transcriptional regulation.

We identified three MRs (*MEF2A*, *TRIM25*, and *RFX2*) with opposite behavior compared to the other ten MRs. *TRIM25* (Tripartite motif-containing 25) works as a modulator of the innate immune response against viruses and limits the inflammatory response by interacting with other genes in signaling cascades, protecting the host [48]. This TF also ubiquitinates stratifin and the caspase-recruitment domain (CARD) of retinoic-acid-inducible gene I (RIG-I). This process negatively regulates the cell-cycle progress and induces type I interferon and NF-κB [49,50]. It was also shown that *TRIM25* is upregulated in organisms challenged with LPS [49], corroborating our results. Type I Interferon and NF-κB increase the expression of pro-inflammatory mediators and extend neutrophil lifespan. It leads to a positive feedback loop of pro-inflammatory molecules and to an exacerbated inflammatory response [51].

*MEF2A* has recently been associated with sepsis progression. In a recent paper, Zhang and collaborators identified *MEF2A* upregulated in whole blood of adult septic patients. According to the authors, the long non-coding RNA MIR155HG facilitates sepsis progression by upregulating MEF2A [52]. The authors also link the same mechanism to apoptosis induction in RAW 264.7 and HL-1 cells treated with LPS [52]. *MEF2A* has essential roles in myocyte development and has already been associated with apoptosis via p38 MAPK signaling [7]. Pon and collaborators also found that MEF2A (along with other MEF2 TFs) regulates apoptosis and cytoskeletal structures [53]. Clark and collaborators found that *Mef2* acts as a primary regulator of innate immunity in flies [54]. Additionally, *MEF2A* also regulates the transcription of glucose receptor GLUT4 in both muscular and adipose tissues, which is thought of as a homologous mechanism of adipose tissue regulation in vertebrates [54]. The *MEF2A*-mediated GLUT4 regulation might involve the Warburg Effect, which can be induced through LPS stimulation [55]. This effect leads to a predominance of aerobic glycolysis over oxidative phosphorylation. However, it is unclear if *MEF2A* expression observed here (in whole blood samples) would reflect systemic physiological alterations. There are reports of ICU admitted septic patients who develop a hypermetabolic stress state, leading to hyperglycemia [56], which might be a compensatory mechanism. Van den Berghe and collaborators applied intensive insulin therapy to maintain normoglycemia which led to less morbidity but no significant impact on mortality [57].

*RFX2* has no described role directly associated with sepsis. The *RFX* gene family members associate with upstream TFs involved with immune cell proliferation, such as *KLF5* [5]. *KLF5* was identified as MR on Pediatric Sepsis 1 and 2 networks (Appendix A). Siegler and collaborators described *RFX* family members as part of a chromatin remodeling machinery. This association is related to the expression of Major Histocompatibility Complex II (MHC-II), along with Class II Major Histocompatibility Complex Transactivator (*CIITA*) [38], also identified as a master regulator of Pediatric Sepsis 1 and 2 networks (Appendix A). This MR is also related to interleukin 5 (*IL-5*) expression, which is also regulated by *GATA3* [58]. Recent studies by Linch and collaborators show that IL-5 might have a protective effect on septic patients [59]. Wong and collaborators found that *CIITA* is downregulated in septic patients in comparison to healthy subjects, corroborating our results [6]. Although *RFX2* promotes *CIITA* expression, the results show that other signaling pathways might be regulating MHC-II genes. One explanation could be the interleukin 10 (IL-10) inhibitory pathway induced by the activator protein 1 (AP-1) dimer. AP-1 is formed by a JUNB and FOSL2 heterodimer and is directly responsible for Th17 lymphocyte differentiation [60], also promoting *IL-10* expression. Both *JUNB* and *FOSL2* were identified as sepsis MR candidates common to RA and MS.

The *GATA3* regulon was enriched by non-coding RNA (ncRNAs) activities. GATA3 is one of the biggest regulons, and most of its genes on both pediatric sepsis networks seem to be related to ncRNAs. Most works focusing on these ncRNAs are related to micro RNAs (miRNAs) regulating pathways crucial in sepsis pathophysiology, such as NF-ĸB, and TNF-α [61]. Several other ncRNAs might impact sepsis progression or outcome, such as long non-coding RNAs (lncRNAs) and circular RNAs (circRNAs), promising the development of new drug targets and biomarkers [62].

Our results give direction on how sepsis master regulators work coordinately in disease progression, especially *MEF2A*, *TRIM25*, and *RFX2*, with potential implications in pediatric sepsis prognosis. Additionally, the presented results shed light on MRs that are more specific to sepsis progression. This three-TF regulating axis might act in three steps: exacerbated inflammation (*TRIM25*) associated with septic cachexia (*MEF2A*) and “fine-tuning” of antigen presentation (*RFX2*) (Figure 7). This three-TF axis has not been directly linked to sepsis in the literature, except by *MEF2A* [52]. Therefore, more studies are needed to better understand how inflammatory response MRs and pediatric sepsis MRs correlate. Further research evaluating the association between regulon activity or the MRS expression and clinical aspects could help elucidate the role of such TFs in pediatric sepsis progression. The lack of studies on sepsis treatments targeting these TFs and their genes opens up new perspectives of investigation for this critical condition.

## Figures and Tables

**Figure 1 biomedicines-09-01297-f001:**
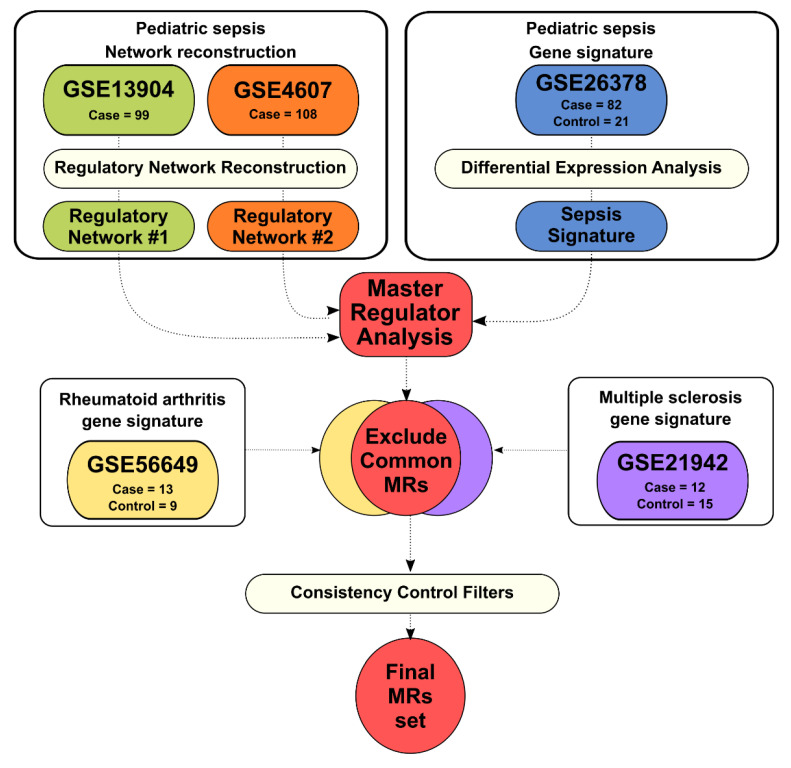
Flowchart of the sepsis regulatory network inference and identification of master regulators. Two independent cohorts were used to infer the networks (GSE13904 and GSE4607) and one cohort was used to build the pediatric sepsis gene signature (GSE26378). The networks were further analyzed by two different gene signatures (GSE56649, rheumatoid arthritis; GSE21942, multiple sclerosis). In the end, a set of 15 master regulators were found: *ZNF529*, *GATA3*, *KLF12*, *RORA*, *HOXB2*, *NR3C2*, *ZNF329*, *RFX2*, *ZKSCAN8*, *ZNF331*, *ZNF235*, *MEF2A*, *ZNF234*, *ZNF134*, *TRIM25*.

**Figure 2 biomedicines-09-01297-f002:**
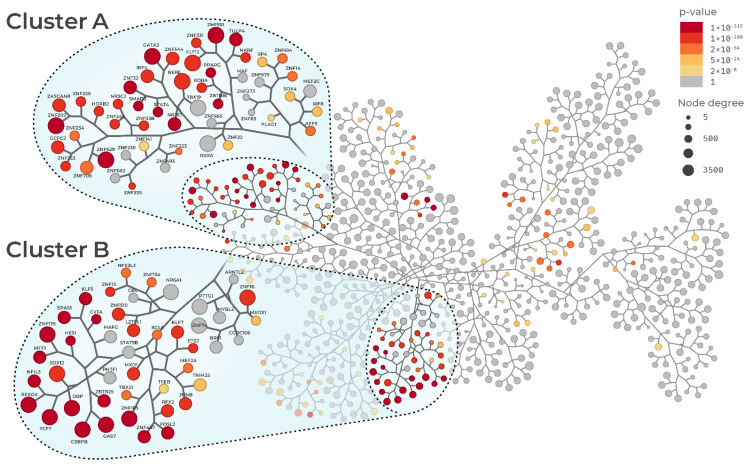
Tree-and-leaf representation of the Pediatric Sepsis 1 (GSE13904) regulatory network. Nodes represent regulatory units (i.e., regulons) labeled after their transcription factors. Node size represents regulon connectivity with regulated genes (degree); node color represents adjusted *p*-value (FDR < 0.01) of MR enrichment for sepsis condition. Dotted lines denote the two main connected clusters of regulons, cluster A and B.

**Figure 3 biomedicines-09-01297-f003:**
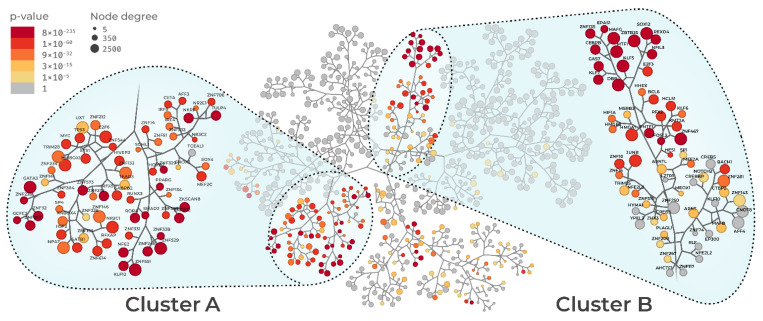
Tree-and-leaf representation of the Pediatric Sepsis 2 (GSE4607) regulatory network. Nodes represent regulatory units labeled after their transcription factors. Node size represents regulon connectivity with regulated genes (degree); node color represents adjusted *p*-value (FDR < 0.01) of MR enrichment for sepsis condition. Dotted lines denote the two main connected clusters of regulons, cluster A and B.

**Figure 4 biomedicines-09-01297-f004:**
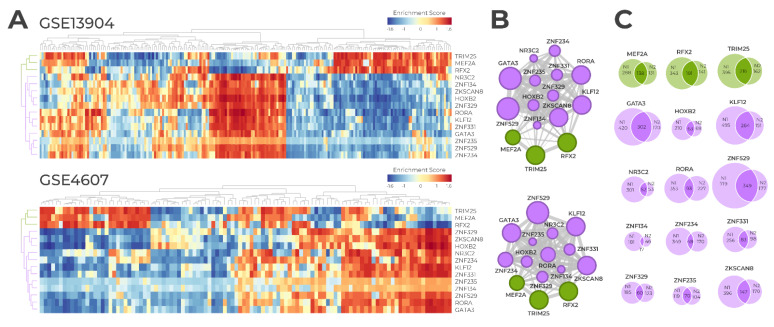
(**A**) Heatmaps showing the regulatory activity of the 15 master regulators in both the Pediatric Sepsis 1 cohort (GSE13904) and Pediatric Sepsis 2 cohort (GSE4607). Y-axis shows the 15 regulons and X-axis shows all samples in each dataset (99 and 108 septic samples in the datasets GSE13904 and GSE4607, respectively). The color code indicates regulon activity, being red when a regulon is activated in a given sample and blue when a regulon is inhibited in a given sample. (**B**) Association map network, showing the connection between the regulons. Nodes represent regulons, and the node widths are proportional to the number of mutual target genes between each two transcription factors. (**C**) Venn diagrams showing the number of genes in each of the 15 regulons, and the regulon similarity comparing Pediatric Sepsis 1 and 2 datasets.

**Figure 5 biomedicines-09-01297-f005:**
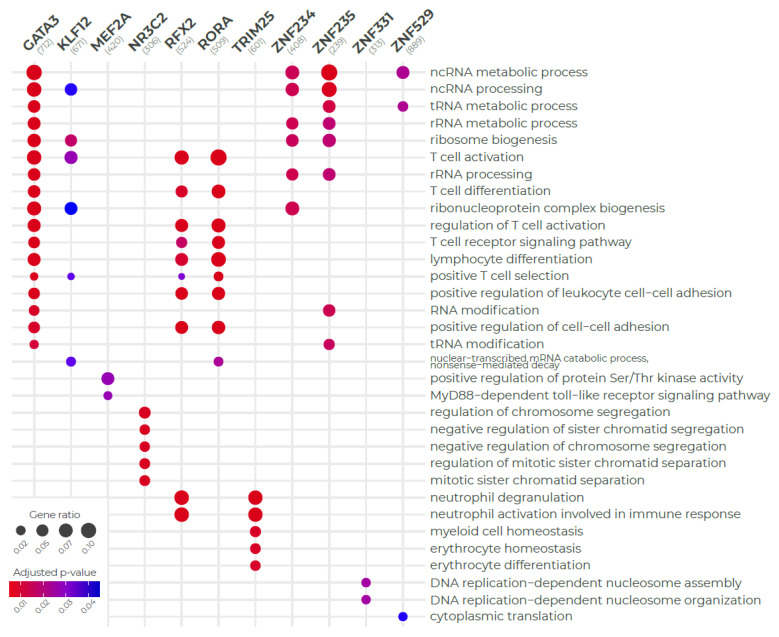
Gene ontology functional enrichment of regulons using clusterProfileR. Each column shows the main biological functions performed by 11 out of the 15 regulons, with a *p*-value < 0.05, adjusted by Benjamini-Hochberg correction. Regulons within this analysis consist of the union of genes regulated by each master regulator in both Pediatric Sepsis 1 and 2 networks.

**Figure 6 biomedicines-09-01297-f006:**
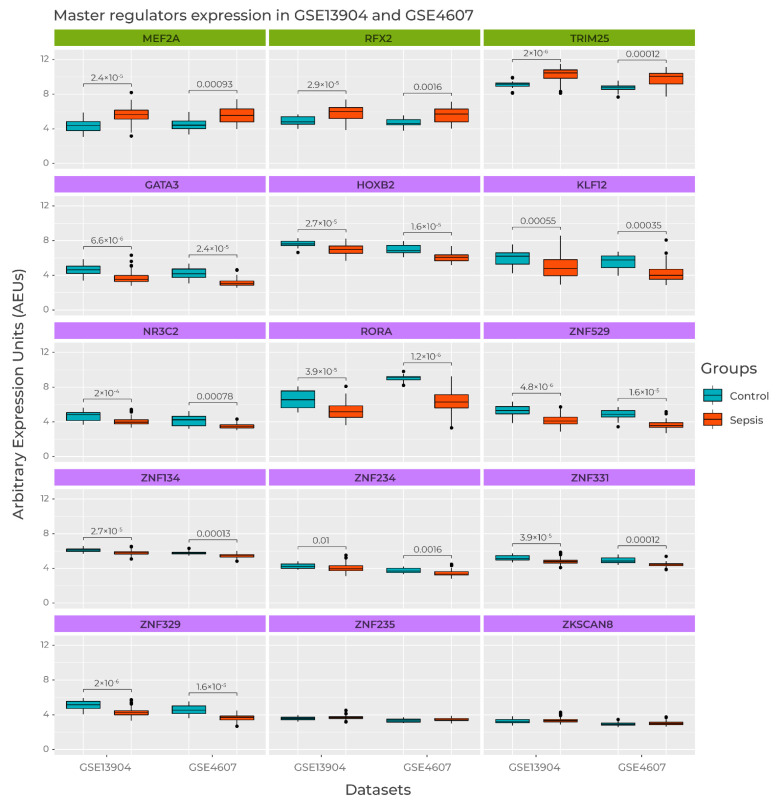
Normalized gene expression of each transcription factor in GSE13904 and GSE4607 datasets. Box heights indicate gene expression in Arbitrary Expression Units; box colors depict septic samples (red), and control samples (blue). An adjusted *p*-value by Wilcoxon paired test was used. Title color depicts an association with high (green) and low (purple) gene expression in both Pediatric Sepsis 1 and 2 datasets.

**Figure 7 biomedicines-09-01297-f007:**
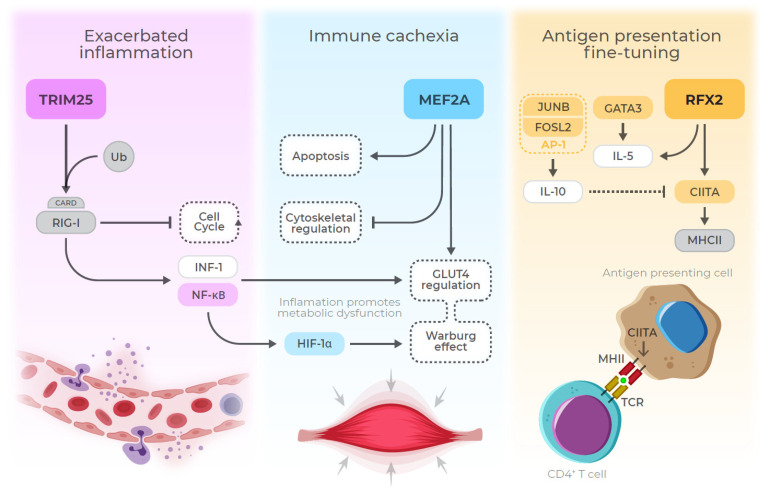
Main findings summary. This figure summarizes the main findings regarding the three top different master regulators: *TRIM25*, *MEF2A*, and *RFX2*.

## Data Availability

Microarray sequenced samples mentioned in this research can be accessed at the Gene Expression Omnibus (https://www.ncbi.nlm.nih.gov/geo/ (accessed on 27 May 2021)) with the accession numbers: GSE13904, GSE4607, GSE26378, GSE56649 and GSE21942. The source code can be found at this repository: https://github.com/dalmolingroup/gene-regulatory-networks.

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
