# Peer review of "Reverse Engineering of the Pediatric Sepsis Regulatory Network and Identification of Master Regulators"

_biomedicines, 2021, doi:10.3390/biomedicines9101297_

Round 1
Reviewer 1 Report
The paper by de Carvalho Oliveir et al describes master regulator in sepsis. The topic is extremely important and research results, although indirectly, they refer to clinical practice.
The manuscript is well written with detailed methodology and extensive discussion. The methods used seem to be original.
Additional data in tables is useful and is well referenced in text. Overall merit of this work is high
Author Response
We thank the Reviewer for the revision and the comments provided.
Reviewer 2 Report
This reviewer is enthusiastic about discovery of two clusters of MRs in healthy and septic shock pediatric patients by de Carvalho Oliveira. The Methods and Results sections are very well written and compelling.
The major problem of this manuscript is the Introduction and to some degree the Discussion. First of all, when putting the work into context, it needs to be clear when the authors refer to adult and when to pediatric sepsis (also applies to the Discussion). Second. I disagree with almost every statement made in the introduction. The sentences are vague ("chemical chaos within cells"???) and often the content incorrect. The literature cited is mostly outdated. The examples start right fromthe beginning:
"Sepsis is a clinical condition characterized by systemic inflammatory response syndrome (SIRS) in addition to a known or suspected infection, commonly in the blood [1]."
No! This refers to the first consensus definition of sepsis (Bone et al. 1992). Here, you are citing Sepsis-3, which defines sepsis as host response to an infection which damages its own organs and tissues.
"A confirmed infection is a bacteremia, viremia, or fungemia, despite sepsis being described to occur in the absence of confirmed infection, for example, induced by lipopolysaccharide [2]."
No! Bacteremia, viremia, or fungemia refer to the presence of bacteria, viruses or fungi in blood. In over half of sepsis case, however, no microbial product is ever detected in blood by microbiology testing and the diagnosis of sepsis largely depends on clinical assessment of the patient. And lipopolysaccharide is used to replicate the initial hyperinflammatory phenotype of sepsis in animals and even humans, or to activate pattern recognition receptors in vitro. But these experimental settings have otherwise nothing to do with sepsis, i.e., we are still looking at a sterile stimulus.
"Since the last decades, sepsis is still the leading cause of death in Intensive Care Units (ICUs) worldwide [3]."
What means since the last decade? I suggest to use a more current report on sepsis epidemiology in the ICU such as Sakr Y et al. Sepsis in Intensive Care Unit Patients: Worldwide Data From the Intensive Care over Nations Audit. Open Forum Infect Dis. 2018;5(12):ofy313. doi: 10.1093/ofid/ofy313. PubMed PMID: 30555852; PubMed Central PMCID: PMCPMC6289022
So on and so forth for the Introduction. I suggest the authors team up with a clinical expert in sepsis and particularly pediatric sepsis to get the context right.
Also in the Discussion, relation to physiology must be clearer. How can one explain regulation of myocyte development in whole blood RNA? And in which cell type do you expect "metabolic dysfunction could be related
394 to NF-ĸB overexpression, which regulates Hypoxia-inducible factor 1α (HIF1α)"?
I hope this helps the authors to identify in which regard their manuscript needs considerable improvement to be publishable. I strongly encourage the authors to invest this additional effort because in my view their reuslts provide significant addition to what can be deduced from whole blood transcriptomics in pediatric sepsis. But please make sure that all parts of the paper are consistent in content (and style) and remove the mistakes in the Introduction and the overall patchwork appearance.
Author Response
## Reviewer 2
Reviewer’s comment: This reviewer is enthusiastic about discovery of two clusters of MRs in healthy and septic shock pediatric patients by de Carvalho Oliveira. The Methods and Results sections are very well written and compelling.
Author’s answer: We thank the Reviewer for the revision and the suggestions. We have addressed all his/her concerns in the revised version.
Reviewer’s comment: The major problem of this manuscript is the Introduction and to some degree the Discussion. First of all, when putting the work into context, it needs to be clear when the authors refer to adult and when to pediatric sepsis (also applies to the Discussion).
Author’s answer: We agree with the Reviewer that it is not clear when we address pediatric sepsis or sepsis in general. We made efforts to point out when we are talking about pediatric sepsis throughout the text in the revised manuscript. We also extensively review the introduction and discussion sections to improve those sections as possible in light of the Reviewer's suggestions and concerns.
Reviewer’s comment: Second. I disagree with almost every statement made in the introduction. The sentences are vague ("chemical chaos within cells"???) and often the content incorrect. The literature cited is mostly outdated. The examples start right from the beginning:
"Sepsis is a clinical condition characterized by systemic inflammatory response syndrome (SIRS) in addition to a known or suspected infection, commonly in the blood [1]."
No! This refers to the first consensus definition of sepsis (Bone et al. 1992). Here, you are citing Sepsis-3, which defines sepsis as host response to an infection which damages its own organs and tissues.
Author’s answer: We thank the Reviewer for the comment. We modified the first introduction paragraph on the revised manuscript. We also withdraw the last sentence of the first paragraph (including “chemical chaos within cells”) since we agree that the sentence was too vague.
Reviewer’s comment: "A confirmed infection is a bacteremia, viremia, or fungemia, despite sepsis being described to occur in the absence of confirmed infection, for example, induced by lipopolysaccharide [2]."
No! Bacteremia, viremia, or fungemia refer to the presence of bacteria, viruses or fungi in blood. In over half of sepsis case, however, no microbial product is ever detected in blood by microbiology testing and the diagnosis of sepsis largely depends on clinical assessment of the patient. And lipopolysaccharide is used to replicate the initial hyperinflammatory phenotype of sepsis in animals and even humans, or to activate pattern recognition receptors in vitro. But these experimental settings have otherwise nothing to do with sepsis, i.e., we are still looking at a sterile stimulus.
Author’s answer: We agree with the Reviewer and withdraw the sentence in the revised manuscript version.
Reviewer’s comment: "Since the last decades, sepsis is still the leading cause of death in Intensive Care Units (ICUs) worldwide [3]."
What means since the last decade? I suggest to use a more current report on sepsis epidemiology in the ICU such as Sakr Y et al. Sepsis in Intensive Care Unit Patients: Worldwide Data From the Intensive Care over Nations Audit. Open Forum Infect Dis. 2018;5(12):ofy313. doi: 10.1093/ofid/ofy313. PubMed PMID: 30555852; PubMed Central PMCID: PMCPMC6289022
Author’s answer: We thank the Reviewer for the reference. We included the suggested reference in the revised manuscript version and modified the first paragraph as suggested.
So on and so forth for the Introduction. I suggest the authors team up with a clinical expert in sepsis and particularly pediatric sepsis to get the context right.
Reviewer’s comment: Also in the Discussion, relation to physiology must be clearer. How can one explain regulation of myocyte development in whole blood RNA? And in which cell type do you expect "metabolic dysfunction could be related to NF-ĸB overexpression, which regulates Hypoxia-inducible factor 1α (HIF1α)"?
Author’s answer: We agree with the Reviewer that we not correctly contextualized in the discussion section the relationship between MEF2A role in myocyte development and sepsis. We rephrased it in the revised manuscript. We carefully revised the introduction and the discussion sections. We withdraw several sentences that we considered vague or confusing in both the introduction and the discussion. We update the reference list by added new references, some of them published after we sent our first manuscript version.
Reviewer’s comment: I hope this helps the authors to identify in which regard their manuscript needs considerable improvement to be publishable. I strongly encourage the authors to invest this additional effort because in my view their results provide significant addition to what can be deduced from whole blood transcriptomics in pediatric sepsis. But please make sure that all parts of the paper are consistent in content (and style) and remove the mistakes in the Introduction and the overall patchwork appearance.
Author’s answer: We thank the Reviewer for the constructive revision provided. In our opinion, it helped to improve the manuscript significantly.
Reviewer 3 Report
The work by Azevedo de Carvalho Oliveira et al. intends to discover transcription factors implicated in the development of sepsis and that may potentially explain the pathophysiology of the disease. For this purpose, the authors used two available gene expression assays of septic patients and performed a series of bioinformatic analyses that revealed several transcription factors as related to sepsis. In particular the authors focused on TRIM25, RFX2, MEF2A, and GATA3.
The study is of interest and brings a new approach to investigate important regulators of sepsis. I have no important comments on this work but it could be of interest to perform associations between the expression levels of these TF and clinical features of the patients if available (clinical scores, disease severity, etc)
Author Response
## Reviewer 3
Reviewer’s comment:
The work by Azevedo de Carvalho Oliveira et al. intends to discover transcription factors implicated in the development of sepsis and that may potentially explain the pathophysiology of the disease. For this purpose, the authors used two available gene expression assays of septic patients and performed a series of bioinformatic analyses that revealed several transcription factors as related to sepsis. In particular the authors focused on TRIM25, RFX2, MEF2A, and GATA3.
The study is of interest and brings a new approach to investigate important regulators of sepsis. I have no important comments on this work but it could be of interest to perform associations between the expression levels of these TF and clinical features of the patients if available (clinical scores, disease severity, etc)
Author’s answer:
We thank the Reviewer for the revision and the suggestion. Unfortunately, the dataset's metadata is limited, and we don't have access to patient's medical records of performing statistical associations robust enough. We agree with the Reviewer that such analysis would be promising. We added a further discussion (last paragraph) to the revised manuscript approaching the necessity to perform additional clinical validation in future investigations.
Round 2
Reviewer 2 Report
The authors have appropriatley addressed all concerns in their revision.
I also appreciate the improve quality of the tables.
Author Response

(The authors gave the same response as above.)
